# Telemonitoring for the Follow-Up of Obstructive Sleep Apnea Patients Treated with CPAP: Accuracy and Impact on Therapy

**DOI:** 10.3390/s22072782

**Published:** 2022-04-05

**Authors:** Cécile Dusart, Stéphanie Andre, Thomas Mettay, Marie Bruyneel

**Affiliations:** 1Department of Pulmonary Medicine, CHU Saint-Pierre, Université Libre de Bruxelles, 1050 Brussels, Belgium; cecile.dusart@slbo.be; 2Department of Pulmonary Medicine, CHU Brugmann, Université Libre de Bruxelles, 1050 Brussels, Belgium; stephanie.andre@chu-brugmann.be (S.A.); thomas.mettay@chu-brugmann.be (T.M.)

**Keywords:** continuous positive airway pressure, obstructive sleep apnea, telemonitoring, leaks, compliance

## Abstract

Continuous positive airway pressure (CPAP) telemonitoring (TMg) has become widely implemented in routine clinical care. Objective measures of CPAP compliance, residual respiratory events, and leaks can be easily monitored, but limitations exist. This review aims to assess the role of TMg in CPAP-treated obstructive sleep apnea (OSA) patients. We report recent data related to the accuracy of parameters measured by CPAP and try to determine the role of TMg in CPAP treatment follow-up, from the perspective of both healthcare professionals and patients. Measurement and accuracy of CPAP-recorded data, clinical management of these data, and impacts of TMg on therapy are reviewed in light of the current literature. Moreover, the crucial questions of who and how to monitor are discussed. TMg is a useful tool to support, fine-tune, adapt, and control both CPAP efficacy and compliance in newly-diagnosed OSA patients. However, clinicians should be aware of the limits of the accuracy of CPAP devices to measure residual respiratory events and leaks and issues such as privacy and cost-effectiveness are still a matter of concern. The best methods to focus our efforts on the patients who need TMg support should be properly defined in future long-term studies.

## 1. Introduction

Obstructive sleep apnea–hypopnea (OSA) syndrome is a highly prevalent disease despite still being under-diagnosed. This disorder is responsible for reduced quality of life (QoL), secondary excessive daytime sleepiness (EDS), negative cognitive and psychological impacts, and contributes to risk of cardiovascular disease, stroke, and diabetes [1]. The severity of the disease is currently based on apnea–hypopnea index (AHI), despite the fact that this definition is poorly correlated with the clinical features and other consequences of the disease, and will be probably adapted in the future [2]. The definition of apnea–hypopnea index (AHI) is number of hypopneas + apnea/hour of sleep (or recording time), measured on a polysomnography (comprehensive sleep assessment, able to assess the exact sleep time) or on a polygraphy (simplified sleep recording where only the recording time is measured).

Continuous positive airway pressure (CPAP) remains the first-line treatment in patients with moderate to severe OSA. Recommendations suggest regular patient follow-up in order to ensure treatment effectiveness, tolerance, and adherence as well as symptom resolution. In this context, telemedicine (TM) is a promising way to optimize care.

The use of TM has increased worldwide during the last decade, particularly during the last two years when its use has been motivated by the COVID-19 pandemic [3]. It is anticipated that the use of TM will continue to grow in the future.

TM can be defined as the use of information and communication technologies to improve patient outcomes by increasing access to care and medical information [4].

Since the development of the first CPAP device, assessment of the efficacy of CPAP for the management of respiratory obstructive events has continuously improved. From clinical and self-reported information, built-in CPAP monitoring systems have been developed to record data as well as respiratory parameters [5], moving monitoring from hospital to home.

Today, TM has a preponderant role in the field of sleep medicine and is important to every stage of management of OSA, from diagnosis to CPAP treatment monitoring. Recording system data allows patient follow-up at home and remote adaptation of some parameters, a practice termed telemonitoring (TMg). Recorded data review is also very helpful during consultations, in addition to clinical assessment, in order to objectively document compliance, treatment efficacy, and the cause of side effects (e.g., leaks).

OSA is one of very few chronic diseases for which it is possible to objectively measure treatment compliance due to the availability of information about daily CPAP device use via TMg.

The purpose of this review is to assess the role of TMg in CPAP-treated OSA patients. We aim to assess recent data related to the accuracy of the parameters measured by CPAP. Accuracy refers to the degree to which a measurement, or an estimate based on measurements, represents the true value of the attribute being measured. This assessment is essential to understand the validity of CPAP reports. We also aim to determine the role of TMg in CPAP treatment follow-up, from the perspective of both the healthcare professional and the patient.

## 2. Telemonitoring of CPAP-Treated Patients

### 2.1. Measurement and Accuracy of CPAP-Recorded Data

CPAP tracking systems are built with connectivity to allow remote access by way of Global System for Mobile Communication (GSM) or General Packing Radio System (GPRS), providing information about daily use, pattern of use, respiratory events (residual AHI), type of residual events (central, obstructive), mask leaks and CPAP pressure. Data are transferred on a daily basis and are available via a central secured data center (cloud) for healthcare providers/professionals or, for some devices, via an application, accessed directly by the patient, to stimulate self-management (Figure 1).

Figure 2A,B show an example of the type of data that can be extracted from TMg as a summary of the data collected from the CPAP device, in patients exhibiting irregular use and residual respiratory events during sleep. The pattern of CPAP use can also be interesting information. For example, in cases of comorbid sleep disorder or when sleep is disturbed due to environmental factors (e.g., shift work, narcolepsy, Ramadan) (Figure 3A,B and Figure 4A,B).

At the current time, there is some variability in the definition of the different parameters mentioned above, as each manufacturer of CPAP uses different systems to measure and collect this data. However, it is essential to clearly understand the different definitions to optimize their interpretation and use.

To begin, we need to know if CPAP devices work. CPAP aims to maintain a positive airway pressure (PAP) during inspiration and expiration. To provide this constant pressure, the flow rate must be adjusted to compensate for the loss of pressure between the flow generator and the patient’s airway caused by leakage and variability in breathing. Sensors placed in CPAP device measure airflow, pressure, and motor speed. Two points of pressure measurement are required to calculate the flow. Flow calculation is also influenced by mask type and tube, so clinicians must manually enter this data into the device. Low and high pass filters are applied to flow in order to exclude artifacts related to coughing or cardiogenic variations. Analysis of the breathing cycle is also provided: the onset of inspiration is determined by the change from negative to positive flow. Inspiration and expiration detection is required to apply expiratory pressure relief (small drops of pressure during expiration, aiming to increase patient’s comfort) and to detect inspiration flow limitation (in auto-adjusting PAP devices) [6].

The definitions and algorithms used for the automatic detection of residual AHI in CPAP are crucial to understanding the collected parameters. When measured by CPAP device, the definition of AHI is completely different from that used for assessment by polysomnography (PSG) [7,8].

In this case, the AHI is used to assess CPAP therapy. Generally, CPAP measured hypopnea and apnea are defined by flow analysis, which is compared to recent flow to see if it remains under a threshold for at least 10 s (<40% or <80% for hypopnea and apnea, respectively). Residual AHI is commonly called AHI “flow” or AHI “PAP” [6]. Different CPAP devices incorporate their own algorithms, that differ from each other, to detect, measure, and integrate events occurring during the night. For example, RESMED uses a root mean square (RMS) [9] while PHILIPS (Respironics) uses weighted peak flow (WPF) [10] to detect and react to respiratory events.

Detection of central apnea–hypopnea (vs. an obstructive event) is based on a forced oscillation technique in the RESMED machine. The RESMED CPAP oscillates the flow when it falls; if oscillations dissipate, it means that the upper airways are open, reflecting a central event. The Philips system is based on a cardiogenic pulse artefact. A pressure pulse is given a few seconds into apnea and, if it is larger than the expected breath at the end of the apnea, it is considered to be obstructive.

The types of residual events provided by the different devices vary by whether or not the device records central hypopneas and inspiratory flow limitation [6].

Understanding the manner by which the presence of residual respiratory events is determined is essential for two reasons:Accurate interpretation of the CPAP data report.In cases where an auto-adjusting PAP (APAP) is used, the machine will react, by gradually increasing or decreasing the pressure, according to the type of respiratory event detected, and clinicians should be aware of the algorithm used by the device in order to be able to correctly interpret any inappropriate reactions of the APAP device during nighttime recordings.

The denominator used to determine AHI is recording time, in contrast to PSG, providing the real total sleep time. The device calculates the recording time as the length of time the device is turned on, associated with a measurable breathing signal [11].

Results from studies of the accuracy of AHI computed by different CPAP devices are rather conflicting. CPAP has been validated against polysomnography (System One Remstar Auto A-Flex, Philips) by Gagnadoux et al. [12] as well as by Li et al. [13]. Li et al. reported AHI overestimation when the manually scored AHI was low and an AHI underestimation when the manually scored AHI was high, as described in previous studies [13]. These weak correlations appear to be due to the device’s hypopnea detection. In a recent study, Fanfulla et al. compared automatic and manual AHI “scorage” under real-life conditions, including 300 OSA patients treated with different PAP therapies (53% on CPAP) [14]. The authors concluded that automatic detection of residual AHI by the built-in software was not reliable in a real-life context, with the results demonstrating a high percentage of false negatives that were not predicted by the type of PAP therapy, mask, level of leaks, or persistent symptoms [14]. Recently, Midelet et al. [15] evaluated variation in reported residual AHI between two different CPAPs for a given patient. They used a database of 3102 patients (69 having changed CPAP brand), and demonstrated significant differences in residual AHI between CPAP brands, including or not central hypopneas and respiratory effort-related arousals (RERAs). The difference in AHI was already observable after a TMg window of 7 days and constant over time [15].

CPAP devices also provide leak measurement. Leaks are measured in different ways according to different CPAP manufacturers (e.g., RESMED: unintentional leaks + mouth leaks, PHILIPS: intentional leaks subtracted from total flow), leading to different definitions of large leak threshold. Intentional leaks are mandatory for C02 removal and depend on mask type (<24 L/min for nasal mask and <36 L/min for naso-buccal mask for RESMED) and pressure level [5]. Leak compensation is foreseen by CPAP devices to maintain target pressure, and is variable from one device to another. Leaks vary from one breathing cycle to another and unexpected leaks can occur (mouth breathing, lack of seal of the mask). An inappropriate leakage will be compensated for by the device, based on mean expected flow in several breathing cycles, but can be less reliable in the case of large leaks, and lead to difficulties in maintaining target pressure [6,16].

### 2.2. Clinical Management of CPAP-Recorded Data

When a patient is telemonitored, CPAP data can be reviewed daily on a secured platform. In order to focus on challenging patients, manufacturers have created filters to select patients with low compliance (generally < 4 h/night), high leak levels (different thresholds are proposed), and increased residual AHI. According to these data, a joint statement of the Société Française de recherche et medicine du sommeil (SFRMS) and the Société de pneumologie de langue française (SPLF) has been recently published that proposes an algorithm to manage compliance, leaks, and residual AHI in telemonitored CPAP-treated OSA patients [17]. In cases of low compliance, a consultation is proposed. In cases of high leak levels, mask fitting should be reviewed, and buccal leaks should be identified. Finally, residual AHI > 10/h should be investigated. In cases involving obstructive events, inappropriate mask fitting, pressure, or nasal obstruction should be examined, and, for central events, a medical consultation is required. The recommendations suggest, depending on the variability of CPAP device algorithms, to manage only patients with an AHI flow > 10/h on 7 consecutive days. AHI flow is only considered if leaks are resolved and compliance is optimal.

Based on a compilation of seven studies, Verbraecken has also proposed an algorithm with a similar cut-off [18]. AHI ≥ 10 should be managed. In cases of obstructive events, attention is focused on the level of pressure, nasal obstruction, and mask type, and in cases involving central events, medical support is required [18]. The number of consecutive days over which the problem must persist is not described, but it is between 2–5 days in the majority of the studies on which this algorithm is based.

### 2.3. Impact of Telemonitoring on CPAP Therapy

#### 2.3.1. Compliance—Adherence

CPAP use is highly dependent on the willingness of the patient to use the device and to apply the mask while sleeping [18]. In order to assess CPAP use, two terms should be distinguished: adherence refers to the real use of CPAP when this treatment has been prescribed (yes/no) while compliance refers to the daily usage expressed in hours (or minutes) per night in persistent users [19].

Many cohort studies have demonstrated the relationship between CPAP and treatment outcomes as improvements in daytime sleepiness [20], and mortality [21], and in terms of decreased cardiovascular events [22]. However, these long-term benefits are observed in a setting of minimal adherence and compliance (compliance defined as use during at least 4 h/night and for more than 70% of nights). A cut-off of 4 h/night is a minimum goal, as the literature suggests a dose–response relationship between CPAP use and outcomes, with different clinical features improved only by better compliance [23,24]. Moreover, three recent large, randomized control trials (RCTs) that aimed to assess the role of CPAP in secondary prevention for cardiovascular events (CVEs) (RICCADSA trial [25], SAVE trial [26], and ISAAC trial [27]) failed to show a positive effect of CPAP. However, in post hoc analysis of data from the RICCADSA and SAVE trials, patients using CPAP for more than 4 h/night had fewer subsequent CVEs. Khan et al. confirmed, in a meta-analysis based on seven RCTs, that CPAP therapy might reduce major adverse CVEs (primary/secondary prevention) when CPAP is used >4 h/night [28].

Despite awareness of the importance of adherence and compliance, and the development of different strategies to improve them, including education sessions, cognitive behavioral approaches, or nurse coaching, results remain disappointing, with adherence already decreasing significantly during the first year of treatment and compliance reported to be as low as 30–60% [29]. Recent studies have shown different patterns of use among countries, with reported 3-month CPAP compliance of 69% and 88% in the United States and Belgium, respectively [30,31]. Regarding adherence, CPAP drop-out rates of 27% and 48% have been observed in Belgium and in France after 3 years [32,33]. These discrepancies highlight the fact that organization of care around CPAP therapy plays an important role in patient adherence and compliance. Indeed, close follow-up of these patients is mandatory, especially at the beginning of treatment, as 38% of patients require mask changes and 7.5% require pressure modifications in the first 6 months of telemonitored CPAP therapy [34]. There is also reason to believe that long-term follow-up should be kept in the hands of specialized professionals. In France, after 16 months of treatment, patients can be followed by general practitioners, with generally less experience in OSA management. This could be one of the explanations for decreased long-term adherence compared to Belgian statistics.

However, several other predictive factors of poor compliance have been identified, such as low CPAP use and/or side effects at 1 month [35], oro-nasal masks, and depression [36]. Compliance at 1 month is associated with compliance at 24 months [37]. The first weeks of treatment are thus a crucial period to establish long-term usage patterns and close follow-up of CPAP beginners is necessary to support them [38].

In this context, TMg appears to be an opportune new strategy to obtain, and then maintain, adherence over time, including early identification of patients at risk of low compliance. Initial RCTs have demonstrated a positive and significant but rather modest effect on compliance with improvement of 191 versus 105 min at 3 months (mean difference 87 min, 95% CI: 25–148) [39] and 298 versus 99 min at 12 months [40].

Recently, 16 RCTs have focused on the role of TM interventions versus usual care (UC) for improvement of short-term compliance in CPAP-treated patients. Twelve of these studies were included in a meta-analysis from Chen et al., showing that TM is associated with an increase in daily use of CPAP of 0.79 h (95% CI: 0.56–1.01) [41]. All 16 RCTs were included in a systematic review from Labarca et al. that showed that TM was associated with a compliance improvement of 29 min (95% CI: 11.8–46.7) [42]. In this study, the risk ratio (RR) of being compliant for >4 h/night was 1.09 (95% CI: 1.02–1.17) for TM versus UC. The authors also evaluated which method of TM delivery was most effective between call-based (six studies), web-based (five studies), and connected device attached to the CPAP (five studies). Call-based intervention was associated with the best improvement in usage per night, +29.73 min (95% CI: 16.2–43.2), and TM with the connected device attached to the CPAP was associated with a RR of being adherent for >4 h/night of 2.06 (95% CI: 1.20–3.54) [42].

Patient engagement can also influence compliance with the addition of further TM interventions to TMg. Malhotra et al. demonstrated improvement in CPAP compliance with TMg + active patient engagement technology via the use of the MyAir application, RESMED [43]. This very large retrospective study, based on 128,037 patients, reported increased average usage to 5.9 h/night versus 4.9 h/night in the UC group, as well as better compliance of struggling patients using TMg compared with UC [43]. Kuna et al. reported on a study in new CPAP-treated patients who were randomly followed with UC, received UC and personal access to an online portal to check CPAP device data, or received this option with the addition of a financial incentive in the first week (up to $30/day for objective CPAP use ≥4 h/day) [44]. Compliance was significantly better in the groups with access to the online portal versus UC (6.3 ± 2.5 h/night with financial incentive, 5.9 ± 2.5 h/night without, and 4.7 ± 3.3 h/night for UC alone). Disappointingly, access to the portal decreased rapidly over time, with fewer than 20% of patients logging in to the portal after 3 months [44].

This study highlights the fact that long-term patient engagement is difficult to maintain with telehealth programs. This phenomenon has also been described in other chronic diseases and in weight-loss maintenance in obese patients [45]. Thus, we should keep in mind that TM interventions can help to increase compliance but are not intended to replace educational sessions and coaching from specialized healthcare professionals in OSA care. Figure 5 illustrates the necessity of a close monitoring following interventions related to CPAP therapy.

#### 2.3.2. Patient-Reported Outcomes (PROs)

Assessment of the impact of CPAP telemonitoring on therapy is often focused on CPAP compliance rather than on other outcomes. However, with regard to EDS, TM interventions were associated with an improvement in the Epworth Sleepiness Score (ESS) in 7 out of 16 studies in the meta-analysis Labarca et al. [42]. The mean difference was low, 0.52 (95% CI 0.12–0.93), and did not reach the minimal clinically important difference (MCID) for ESS, which lies between 2 and 3 [46]. This can be explained by the poor performance of the ESS [2], or by the inclusion of non-sleepy patients in seven studies, who generally exhibited an improvement of only 1 point on the ESS with CPAP [47]. Indeed, ESS at baseline was not associated with adherence > 4 h/night. It should also be noted that, when comparing CPAP + TM versus CPAP alone, it is clear that, due to the limited increased in compliance obtained with TM, the subsequent difference in EDS is insignificant.

In a study from Sparrow et al., in addition to compliance improvement, the authors described an association with improved functional status, assessed by the Functional Outcomes of Sleep Questionnaire and the Sleep Symptoms Checklist, and fewer depressive symptoms (Center for Epidemiological Studies Depression Scale) [40]. However, due to very low CPAP compliance in the two arms of the study (2.4 h/night in the TM arm vs. 1.5 h/night in the UC arm (mean difference 87 min, 95% CI: 25–148), the importance of these findings is difficult to interpret.

Few studies have aimed to assess QoL. However, QoL was demonstrated to be improved with multimodal TMg by Pépin et al. in a randomized trial performed on 306 OSA patients [48]. Previous reports were negative [49,50,51,52].

#### 2.3.3. Cardiac Events: Interpretation of CPAP-Detected Central Sleep Apnea and Cheynes-Stokes Respiration

Particular attention should be given to patients with residual central events, as Prigent et al. have shown that the detection of incident Cheynes-Stokes respiration by CPAP telemonitoring is associated with a high prevalence of heart failure (31%) and arrhythmias (8%) based on a study of 555 patients followed during a one-year period [53]. This highlights another benefit of TMg, an increased ability to consider, detect, and manage cardiovascular comorbidities.

#### 2.3.4. Integrated Care for OSA and Associated Comorbidities, including Telemonitored CPAP

OSA patients, mainly males, exhibit a high prevalence of comorbidities, with half of them suffering from cardiovascular disorders and about 20–30% from diabetes and dyslipidemia [54]. Comorbidities can either be a causal factor of OSA (e.g., diabetes, heart failure) or a consequence (e.g., stroke, hypertension). CPAP is one part of the overall treatment in these patients, that aims to reduce/control cardiometabolic risk. Integrated strategies using telehealth have been studied in OSA. For example, Pépin et al. studied 306 patients suffering from both OSA and high cardiovascular risk [48]. Patients were randomized to CPAP or to multimodal TMg. In the TMg arm, self-measured morning and evening blood pressure (BP) and physical activity were measured by connected devices. The blood pressure monitor and the actigraph (collecting number of steps/day and sleep periods) were connected to a secure website, accessible for physicians and home care providers. CPAP adherence, leaks, and residual events were collected via TMg. Symptoms and quality of life were assessed using a questionnaire-based application. Multimodal TMg provided predefined interventions to home care providers (managing leaks, mask problems, or other side effects) while the medical team managed residual events or CPAP lack of efficacy. After 6 months, no differences in BP reduction or in physical activity were observed, but CPAP compliance (5.28 ± 2.23 vs. 4.75 ± 2.5 h/night), sleepiness, and QoL were significantly improved in the multimodal TMg arm [48]. Further studies need to define the best way to provide a holistic approach to these patients, including TM tools.

### 2.4. Telemonitored CPAP Therapy: Patient Perspectives

From the patient’s perspective, TMg is often perceived as a mean of reassurance [55]. In several RCTs, satisfaction rates were better in the TM arm using different tools (e.g., feedback by phone, TMg, televisits) compared to the UC group [56,57] or equivalent [58]. However, it is important to fulfill some prerequisites when starting TMg in patients. First of all, patient education about TM methods is important to assure that patients understand the method and are suitable candidates for TM. TM interventions have the theoretical advantage of allowing elderly people or patients living far away to avoid visits to the hospital; however, in real-life settings, the use of TM can be limited by technological barriers/fear, especially if it requires patient intervention to connect to healthcare providers (e.g., for videoconsultation). Costs billed to the patient can also be a source of worry. When restricted to remote TMg for CPAP-treated OSA patients, no special manipulation is required to allow data monitoring, such that there are virtually no limitations.

Considering concerns about privacy protection is also essential. Indeed, despite expressing satisfaction and agreement with the usefulness of online information, some patients have reported concerns about privacy, judging TMg to be intrusive [59]. Bros et al. reported the same concerns, with a majority of patients finding TMg useful but 40% considering TMg to be intrusive [59]. However, in a recent study from Carlier et al. related to interventions performed in telemonitored CPAP patients during the first 6 months of treatment, acceptance (obtained via informed consent) was as high as 87% [34]. Similar data (acceptance of 78%) was also reported in another study [59]. In the latter study, men had fewer concerns about TMg than women, and non-working people were more favorable toward TMg than active people [59].

In addition, acceptance of TMg seems to be an additional predictive factor of adherence. Bros et al. showed that patients refusing TMg were more at risk of CPAP withdrawal [59]. CPAP discontinuation was also more frequent in cases of TMg interruption.

Finally, TM has entered the daily life of many family households through the growing use of connected monitoring tools (e.g., apps, watches, accelerometers) that collect physiological parameters such as blood pressure, sleep measurements, physical activity, and weight. In the future, all of these parameters, including those concerning CPAP treatment, could be integrated into a more comprehensive health monitoring strategy [18]. However, healthcare professionals should be careful to avoid dehumanization of care through TM. Increased use of technology could limit patient engagement if no attention is dedicated to human relationships. The first wave of the COVID pandemic was a good example of this drift, which was experienced as very negative for COVID patients but also for a large part of the general population who experienced limited access to general practitioners, increased use of teleconsultations, and delays in obtaining medical care. Other concerns have also been described in the context of telehealth in COPD patients: it can create high dependency for patients, increase the frequency of nurse–patient interactions, and finally be counterproductive, leading to overtreatment and overmedicalization of patients [60].

### 2.5. Telemonitored CPAP Therapy: Healthcare Professional Perspectives

Many advantages of the use of TM by medical and paramedical providers are expected or have already been demonstrated. Considering compliance problems and the relationship of these problems to the initial days of use, TMg is a good strategy for identifying patients struggling with their therapy, and setting up intensive early interventions for selected patients. Hoet et al. demonstrated that TMg reduces delay to first technical intervention in CPAP-treated patients, a factor that was associated with improved compliance at 3 months [47]. In this way, TMg also allows practitioners to avoid losing time in unnecessary nurse visits or medical follow-up with patients who do not need it [61,62]. TMg could, therefore, help to forge a compromise between increasing needs in sleep disorder diagnosis and support, and limited health care resources. For example, Anttalainen et al. demonstrated, in 111 patients, a savings of 19 min in nursing time between TMg patients (39 min, range 12–132) and the UC group (58 min, range 40–180) [61]. Reduced staff time was also documented by Munafo et al., who showed that the number of contacts per patient was reduced in the telehealth group (TMg) vs. UC: 2.2 ± 2.6/patient vs. 7.8 ± 4.1/patient [62]. Time spent per patient by the staff was also significantly reduced, even in non-compliant patients. Finally, TMg also allows remote monitoring in patients living in medical desert areas, increasing healthcare accessibility [63].

However, two crucial questions remain unanswered: who to monitor? how to monitor?

#### 2.5.1. Who to Monitor?

RCTs on TMg have focused on newly-diagnosed OSA patients for obvious reasons. Precocious adherence is associated with long-term adherence [64], and efforts should be made at the start of the therapy to support and educate patients, and to resolve side effects related to CPAP use. A rapid analysis of patient problems through TMg can help obtain prompt action via remote troubleshooting. Indeed, it has been documented that the majority of treatment adaptations (e.g., mask changes, pressure adjustments) appear in the first few months of treatment. For example, it has been reported that 7% of patients need a mask change and 9% need humidification in the first month of treatment [56], and these needs rise to 43% and 35%, respectively, over the first 3 months [47]. The right patient to monitor is, thus, the new CPAP starter. However, the appropriate duration of TMg remains unclear. In our opinion, TMg should probably last 2 years, as mask change rates and CPAP treatment adaptation decrease drastically after 2–2.5 years (personal experience). Compliance is also expected to improve each year by 8 min/night/year in persisting users [64]. If patients are compliant after 2 years, with an adequate mask and respiratory events that are well controlled after this period, we can be confident that the patient will be able to correctly continue the therapy.

Other target groups can also benefit from TMg:-Long-term users with unexpected high residual respiratory events detected at consultation: TMg can help to monitor whether the problem is improving (or not);-Long-term users who become non-compliant: focusing on treatment and giving more support and feed-back for a limited period can help the patient to become a regular user again.

Prior to TMg, informed consent must be signed by the patient after a comprehensive explanation of the program, including a warning that TMg does not release the patient from the responsibility to alert healthcare professionals in case of problems related to CPAP use.

#### 2.5.2. How to Monitor?

When TMg is used, dedicated medical and nursing staff can use it two ways:-Pro-active analysis of TMg data, focused on troubleshooting problems identified by the filters. The frequency of this monitoring can be chosen by the staff (e.g., 1–2 times/week). In previous studies, the frequency of monitoring was highly variable: 1 X/day, 1 X/2 days, 3 times/week, every 6 days [47,56,57,61,62]. If we refer to the proposals from SFRMS/SPLF [17], AHI > 10 should be managed if it persists more than 7 days. According to Verbraecken’s review, alerts should be managed if they persist for 2–5 days [18]. According to these recommendations, weekly management of TMg alerts seems to be optimal. However, this workflow is time-consuming and needs to be adequately financed.-Passive use: TMg data are only reviewed when the patient calls to report a problem, or when the patient comes for a nursing or medical consultation. This organization of care is more acceptable in terms of time dedicated to the patient, but is a misuse of TMg. Many patients with inadequate efficacy of CPAP will not be detected and long delays can occur before detection of a significant problem or non-compliance, which can lead to medical complications (e.g., in case of Cheynes–Stokes breathing) or symptom recurrence and subsequent risks.

### 2.6. Telemonitored CPAP Therapy: Real Costs—Cost-Effectiveness

Few studies have focused on the cost-effectiveness of TMg. Turino et al. have analyzed the treatment and follow-up costs (direct and indirect) in a cohort of 100 TMg-followed CPAP patients [57]. Clinical outcomes were similar in both groups. Three-month costs were lower in the TMg arm vs. UC: 124 Euro versus 171 Euro. In a previous study, Isetta et al. calculated the cost-effectiveness of a TM program versus UC in 139 newly-diagnosed OSA patients [58]. At 6 months, clinical outcomes were similar in both groups and lower costs were billed in the TM arm. When considering medical visits, time to travel to the hospital, and time out of work, the mean cost was 168 Euro in the TM arm and 180 Euro in the control arm, despite the higher number of extra visits to nurses and physicians in the TM arm. The impact of TMg on cost effectiveness seems to be positive but it is not yet clear due to the paucity of the current data [63].

The direct cost of TMg is still very variable from one country to another and from one hospital to another, according to provider policy and to local healthcare reimbursement policies (authorities still lack reimbursement of such tools in many countries). What should really be included in the provider’s billing remains unclear at this time: the access to the cloud? access to the stored data? price per access or all-inclusive price? software maintenance of the cloud or platform? guarantee that the system will be supported for several years?

More than TMg billing, discussion of the generalization of a fee for healthcare professionals managing data and alerts of telemonitored patients should also be started, as this represents additional work for sleep professionals, and is not yet covered by healthcare insurance.

### 2.7. Issues Related to TMg Use

Privacy and data ownership remain two matters of concern when using TMg. Each device comes from one manufacturer, and each manufacturer has their unique CPAP platform and host for storing data in the cloud. This presents the possible risk of fraudulent data access to data stored for millions of users, and the legal limits surrounding TMg use must be well thought out. Another problem is that the users are completely dependent on the manufacturer for access to the data stored in CPAP devices. In the past, universal devices were available to obtain the data from CPAP machines, independently from the manufacturer [47]. Unfortunately, these devices are not yet supported by the manufacturer and dependency on manufacturers for data access has increased.

## 3. Conclusions

The telemedicine revolution is ongoing in many fields of healthcare. TMg for CPAP follow-up treatment monitoring is very simple to use, as it does not require any intervention of the patient to transmit physiological or device-collected data, in contrast to its use in acute or chronic conditions such as COPD or heart failure, or in COVID patients after hospital discharge.

TM appears to be effective for increasing CPAP compliance and should focus on challenging patients and those just starting CPAP. Patient acceptance and satisfaction with TM appear to be positive as well.

TM is easy to use (but is highly dependent on the manufacturer) and there can be a temptation to rapidly transfer CPAP follow-up to less qualified healthcare workers (non-somnologists, general practitioners). The role of the somnologist is to emphasize that beyond CPAP-collected data, there is a patient with potentially sleep-related complaints, comorbidities, and other conditions that should be properly controlled before being sure that close management of the patient is no longer mandatory.

Beyond these aspects, this review has highlighted the limits of TMg and TM tools. First of all, the accuracy of CPAP measured data is heterogeneous, for both leaks and residual respiratory events. TM tools explored in different studies, including TMg, are also very diverse and cannot be taken as a whole, even if TMg is included in the majority of recent studies. Furthermore, we do not have long-term studies on TMg use, making it difficult to determine how long TMg should be carried out. Monitoring schemes (e.g., daily, weekly) also remain a matter of debate. In addition, few studies have focused on the cost-effectiveness of TMg, and privacy issues remain. Finally, using TMg puts the clinician into the uncomfortable situation of being highly dependent on the device manufacturer and software platform provider for access to patient data.

To conclude, use of telemonitoring in newly CPAP-treated OSA patients is an interesting, supportive, and informative tool, but users should be aware of the technological limits, caveats, and related unresolved issues associated with its use.

## Figures and Tables

**Figure 1 sensors-22-02782-f001:**
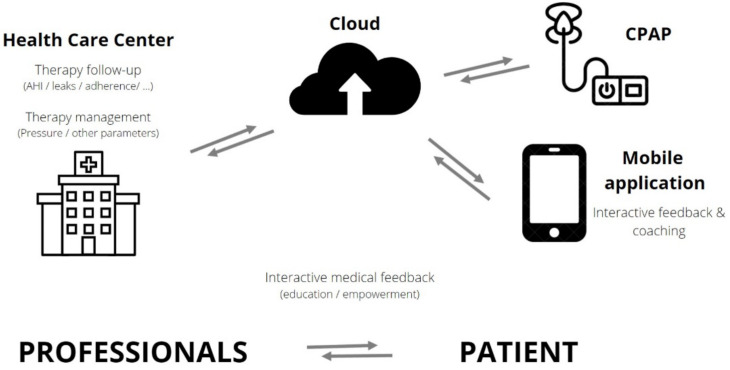
Interactions between patients and healthcare professionals in telemonitored CPAP-treated OSA patients. OSA: obstructive sleep apnea–hypopnea; CPAP: continuous positive airway pressure; AHI: apnea–hypopnea index.

**Figure 2 sensors-22-02782-f002:**
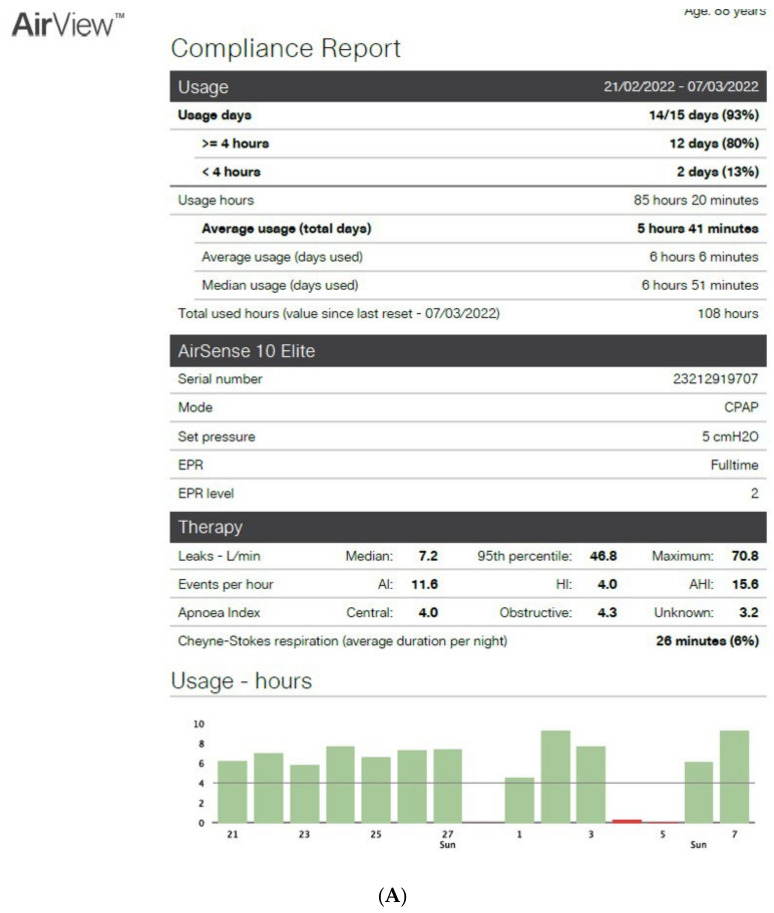
CPAP device data collection ((**A**): Resmed; (**B**): Philips) over an 8-to-15-day period in patients exhibiting irregular use and residual respiratory events during sleep. AI: apnea index, HI: hypopnea index, AHI: apnea–hypopnea index, Avg: average, CA: central apnea, OA: obstructive apnea. Red circles highlight average usage and average residual AHI.

**Figure 3 sensors-22-02782-f003:**
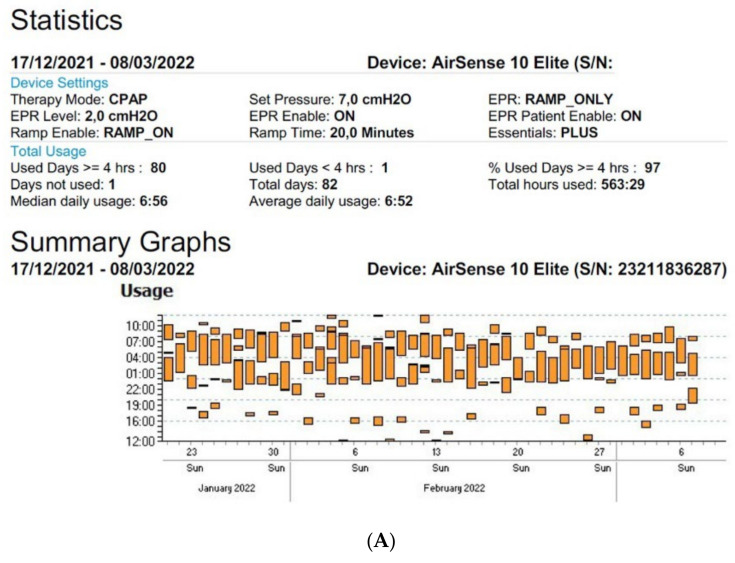
Sleep patterns of CPAP-treated patients suffering from maintenance insomnia. Frequent naps are also observed. CPAP data are collected from Resmed (**A**) and Philips (**B**) devices. AHI: apnea–hypopnea index, Avg: average, CA: central apnea, OA: obstructive apnea.

**Figure 4 sensors-22-02782-f004:**
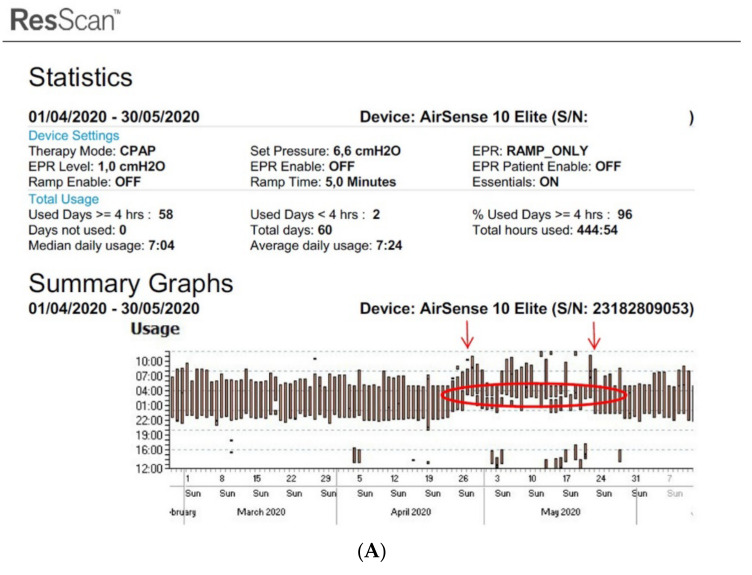
Disturbed sleep pattern of CPAP users during the Ramadan period (arrows). Volitional awakenings, to pray, are observed (**A**). CPAP data are collected from Resmed (**A**) and Philips (**B**) devices. EPR: expiratory pressure relief.

**Figure 5 sensors-22-02782-f005:**
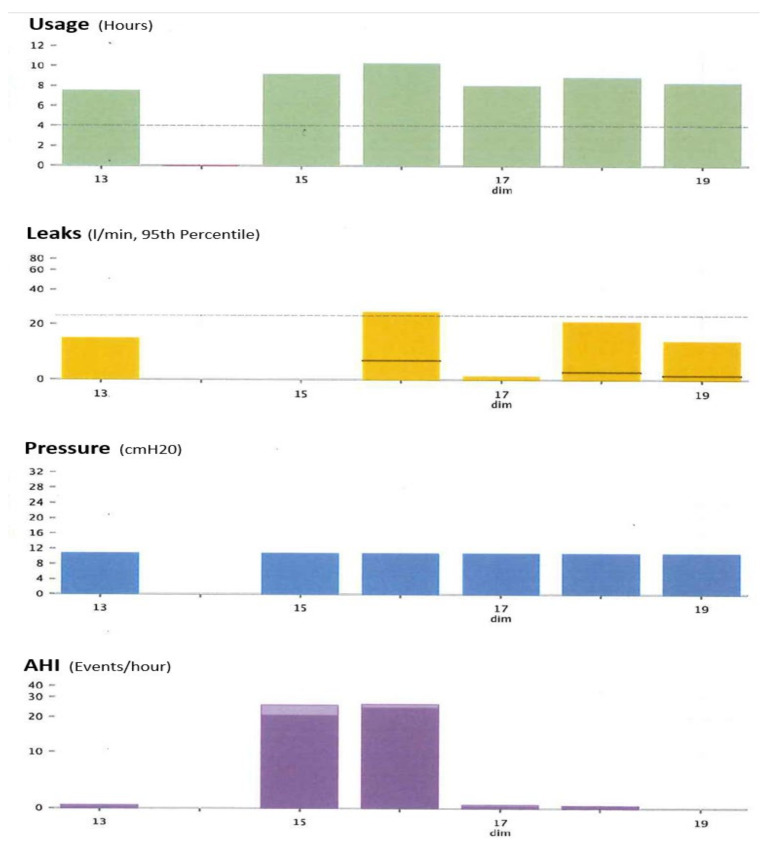
Illustration of the immediate feed-back obtained after mask change. A significant increase in AHI is observed during two nights spent with a naso-buccal mask. Leaks were also significantly increased during the second night. The patient went back to nasal mask use afterwards. AHI: apnea–hypopnea index.

## Data Availability

The datasets used and analyzed during the current study are available from the corresponding author on reasonable request.

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
