# Peer review of "Telemonitoring for the Follow-Up of Obstructive Sleep Apnea Patients Treated with CPAP: Accuracy and Impact on Therapy"

_sensors, 2022, doi:10.3390/s22072782_

Round 1

Reviewer 1 Report

The definition of AHI is completely different from that used for assessment by polysomnography (PSG). The AHI is used to assess CPAP therapy. 

Suggestion: added some representabtive AHI assess studies,

 e.g. weareable PPG or ECG, 

Quantitative detection of sleep apnea with wearable watch device PLOS ONE 2020

AUTOMATED SLEEP APNEA ASSESSMENT BASED ON MACHINE LEARNING AND WEARABLE TECHNOLOGY, SLEEP 2020

deeping learning method, 

A dual-model deep learning method for sleep apnea detection based on representation learning and temporal dependence  NEUROCOMPUTING 2022

Contribution of Different Subbands of ECG in Sleep Apnea Detection Evaluated Using Filter Bank Decomposition and a Convolutional Neural Network, SENSORS 2022

FENet: A Frequency Extraction Network for Obstructive Sleep Apnea Detection IEEE JOURNAL OF BIOMEDICAL AND HEALTH 2021

Author Response

Thank you for this suggestion, we have added to references to document how apnea can be detected.

Reviewer 2 Report

The paper reviews the relevance of the remote feedback that CPAP machines give on the unfolding of the OSA treatment. The article is generally well-written and convincing; however, there are some elements that the authors need to improve, as presented below. 

1. The paper emphasizes (starting with the title) the reliability of measurements made with CPAP machines. However, the authors follow the misinterpretation of reliability that appears in many medical journals. Reliability R(t, t0) is the probability that a system executes its intended function (without failure) at moment t, given that we know that was true at t0. As such, reliability measures the correct operation of the system in the presence of *random* errors. What the authors call reliability looks more like accuracy (which assumes that the system is affected by systematic errors). From the elements presented in the paper, we believe that the authors are instead aiming at discussing the accuracy/precision interplay. Consequently, we recommend that the authors clarify these aspects in formal terms; an engineering journal emphasizes rigor.

2. On line 303, the authors mention measuring the "physical activity" with "connected devices." What are these devices, and what kind of physiological signals are they measuring?

3. The authors intensely use AHI throughout the paper—and for a good reason—yet they do not define it with clarity; the literature explains AHI formally, and the authors should do the same.  

4. Sensors is an engineering/technology-oriented journal. As such, we believe that the authors should briefly review and explain (in a dedicated section or subsection) what physiological signals (e.g., breathing rate, heart rate, breathing pressure) are measured with CPAP devices and how these are translated to assessing, for example, the AHI. 

5. Fig. 4 contains text in French. While some of us have no problem understanding French, which is a beautiful language, please remember that our readership is 100% English-speaking. Indeed, in science and technology, the lingua franca is English. We kindly ask the authors to fix this problem.

6. ResMed and Philips are the leading players in the CPAP market. In Figures 2-3, the authors present some reports generated by ResMed machines, yet they do not specify this in the caption. Moreover, since they discuss the difference between Philips and ResMed CPAP machines, we recommend presenting a similar report generated by a Philips CPAP.  

7. The paper is well-written. Still, it contains some minor language inaccuracies, such as not putting a comma after "e.g." For instance, it should be "(e.g., France, Belgium)."  

Author Response

  1. The paper emphasizes (starting with the title) the reliability of measurements made with CPAP machines. However, the authors follow the misinterpretation of reliability that appears in many medical journals. Reliability R(t, t0) is the probability that a system executes its intended function (without failure) at moment t, given that we know that was true at t0. As such, reliability measures the correct operation of the system in the presence of *random* errors. What the authors call reliability looks more like accuracy (which assumes that the system is affected by systematic errors). From the elements presented in the paper, we believe that the authors are instead aiming at discussing the accuracy/precision interplay. Consequently, we recommend that the authors clarify these aspects in formal terms; an engineering journal emphasizes rigor.

Thank you for this excellent comment, we have changed "reliability" by "accuracy" throughout the manuscript and have defined "accuracy" in the introduction.

  1. On line 303, the authors mention measuring the "physical activity" with "connected devices." What are these devices, and what kind of physiological signals are they measuring? Details related to monitoring tools have been added.

3.The authors intensely use AHI throughout the paper—and for a good reason—yet they do not define it with clarity; the literature explains AHI formally, and the authors should do the same.  Definition has been added in the beginning of “Introduction”.

The definition of apnea-hypopnea index (AHI) is number of hypopneas + apnea/hour of sleep (or recording time), measured on a polysomnography (comprehensive sleep assessment, able to assess the exact sleep time) or on a polygraphy (simplified sleep recording where only the recording time is measured).

  1. Sensors is an engineering/technology-oriented journal. As such, we believe that the authors should briefly review and explain (in a dedicated section or subsection) what physiological signals (e.g., breathing rate, heart rate, breathing pressure) are measured with CPAP devices and how these are translated to assessing, for example, the AHI. 

Thank you for this comment. A paragraph aiming to describe measurements of physiological signals by CPAP has been added, Chapter 2.1. Measurement and reliability of CPAP-recorded data.

  1. Fig. 4 contains text in French. While some of us have no problem understanding French, which is a beautiful language, please remember that our readership is 100% English-speaking. Indeed, in science and technology, the lingua franca is English. We kindly ask the authors to fix this problem.

Sorry for the inconvenience. The figure has been adapted.

  1. ResMed and Philips are the leading players in the CPAP market. In Figures 2-3, the authors present some reports generated by ResMed machines, yet they do not specify this in the caption. Moreover, since they discuss the difference between Philips and ResMed CPAP machines, we recommend presenting a similar report generated by a Philips CPAP.  

Thank you for this interesting suggestion, we have added similar reports from Philips devices for Figures 2,3 and 4.

  1. The paper is well-written. Still, it contains some minor language inaccuracies, such as not putting a comma after "e.g." For instance, it should be "(e.g., France, Belgium)."  

Sorry for these mistakes. We have made the corrections throughout the manuscript.

Reviewer 3 Report

This is an interesting and timely review of the telemonitoring of CPAP use given the development in the relevant field and increase in its use in patient management. I have noted below some points that I believe would help improve this manuscript.

Line #s 61-64: This section seems irrelevant to the focus of this manuscript.

Line #s 101-115: References are needed to support the statements given in this section.

Line #s 182-185: References are needed to support these definitions.

Line # 236: Good to have a confidence interval for 29.73 min if available.

Line #s 243, 249-250: provide confidence intervals for compliance values if available.

Line # 254-255: It is difficult to say ‘never’ scientifically and without evidence to that. Suggest toning down this statement so as not to negate any distant or future possibilities.

Figure 4: Must indicate what IAH [not AHI] means.

Lack of confidence intervals for stated compliance or improved outcomes is commonly seen throughout this manuscript. I suggest that the authors indicate these in all such instances, as this is important information to judge the usefulness of any such improvements.

Line #s 332-333: Does the high acceptance rates necessarily mean a lack of concerns? Are there reports of subsequent problems even after the initial acceptance of TMg?

The discussion could be more aptly named 'conclusions' as the discussion is already embedded in the rest of the review.

Author Response

Line #s 61-64: This section seems irrelevant to the focus of this manuscript.

According to your suggestion, we have deleted these sentences.

Line #s 101-115: References are needed to support the statements given in this section.

References were indeed essential, they have been added.

Line #s 182-185: References are needed to support these definitions.

One reference was also added.

Line # 236: Good to have a confidence interval for 29.73 min if available.

CI has been added.

Line #s 243, 249-250: provide confidence intervals for compliance values if available.

For the data of line 243, CI is unfortunately not given in the paper of Malhotra et al.

Line #s 249-250: mean+ standard deviation is now provided

Line # 254-255: It is difficult to say ‘never’ scientifically and without evidence to that. Suggest toning down this statement so as not to negate any distant or future possibilities.

Effectively, the term was poorly chosen. The sentence has been adapted.

Figure 4: Must indicate what IAH [not AHI] means.

As requested by another reviewer, French text was suppressed from Figure 4 (now Figure 5) such that the problem has disappeared.

Lack of confidence intervals for stated compliance or improved outcomes is commonly seen throughout this manuscript. I suggest that the authors indicate these in all such instances, as this is important information to judge the usefulness of any such improvements.

We agree that CI and mean+/-standard deviation were missing for a lot of data, they have been added throughout the manuscript

Line #s 332-333: Does the high acceptance rates necessarily mean a lack of concerns?

There are concerns associated with TMg and telehealth in general. We have add exemples of concerns in COPD patients (+ reference), Chapter 2.4. Telemonitored CPAP therapy: Patient perspectives

Are there reports of subsequent problems even after the initial acceptance of TMg?

We did not find such reports.

The discussion could be more aptly named 'conclusions' as the discussion is already embedded in the rest of the review.

We completely agree, and have followed your suggestion.

Round 2

Reviewer 1 Report

ACCEPT in present form

Reviewer 2 Report

The authors addressed all my concerns in the revised manuscript.